# Towards a conceptual framework for the prevention of gambling-related harms: Findings from a scoping review

**Jamie Wheaton**[1]*, **Ben Ford**[1,2,3], **Agnes Nairn**[4‡], **Sharon Collard**[1‡]

**1** School of Geographical Sciences, University of Bristol, Bristol, United Kingdom, **2** Psychological Sciences, School of Natural and Social Sciences, University of Gloucestershire, Cheltenham, United Kingdom, **3** The Department of Psychology, Edge Hill University, Ormskirk, United Kingdom, **4** University of Bristol Business School, University of Bristol, Bristol, United Kingdom

☉ These authors contributed equally to this work.
‡ AN and SC also contributed equally to this work.
* jamie.wheaton@bristol.ac.uk

## Abstract

The global gambling sector has grown significantly over recent years due to liberal deregulation and digital transformation. Likewise, concerns around gambling-related harms—experienced by individuals, their families, their local communities or societies—have also developed, with growing calls that they should be addressed by a public health approach. A public health approach towards gambling-related harms requires a multifaceted strategy, comprising initiatives promoting health protection, harm minimization and health surveillance across different strata of society. However, there is little research exploring how a public health approach to gambling-related harms can learn from similar approaches to other potentially harmful but legal sectors such as the alcohol sector, the tobacco sector, and the high in fat, salt and sugar product sector. Therefore, this paper presents a conceptual framework that was developed following a scoping review of public health approaches towards the above sectors. Specifically, we synthesize strategies from each sector to develop an overarching set of public health goals and strategies which—when interlinked and incorporated with a socio-ecological model—can be deployed by a range of stakeholders, including academics and treatment providers, to minimise gambling-related harms. We demonstrate the significance of the conceptual framework by highlighting its use in mapping initiatives as well as unifying stakeholders towards the minimization of gambling-related harms, and the protection of communities and societies alike.

## Introduction

The gambling sector has seen significant growth in recent years due to liberal deregulation and digital transformation [1]. As of 2021, the global online gambling industry alone was worth US $61.5 billion, forecast to rise to US$114.4 billion by 2028 [2]. The increasing accessibility of gambling—such as the products available through smartphones [3]—increases the possibility

**Data Availability Statement:** All files are available at the Open Science Framework: https://osf.io/d7js2/.

**Funding:** JW, BF, AN, and SC carried out this work as part of the Bristol Hub for Gambling Harms Research, which is funded by a grant from the national charity GambleAware. GambleAware is funded by voluntary donations from the gambling industry. Governance procedures and due diligence provide safeguards to ensure the Hub's independence from GambleAware and the gambling industry. The funders had no role in study design, data collection and analysis, decision to publish, or preparation of the manuscript. https://www.begambleaware.org/about-us.

**Competing interests:** All the co-authors (JW, BF, AN and SC) have received funding from GambleAware through their work at the Bristol Hub for Gambling Harms Research. The Bristol Hub for Gambling Harms Research (2022-2027) is funded by GambleAware which is funded by voluntary donations from the gambling industry to build capacity in interdisciplinary gambling harms research. Governance procedures and due diligence provide safeguards to ensure the Hub's independence from GambleAware and the gambling industry. Neither GambleAware nor the gambling industry have any input to the strategic, operational or research activities of the Hub. SC has also received research funding from the Gambling Commission Regulatory Settlement funds. This does not alter our adherence to PLOS ONE policies on sharing data and materials.

of gambling-related harms (GRH) [4]. These harms are wide-ranging, covering numerous dimensions (such as financial, emotional or cultural) and they are not restricted to the gambler, also affecting their families and social networks [5]. There is therefore a growing support for a public health (PH) approach to GRH [4, 6–9]. Thomas et al. [7] highlight five key pillars of a PH approach to GRH [7]: the development and implementation of a comprehensive public health framework to *prevent* gambling harm; the elimination of industry influence from research policy and practice; the addressing of structural characteristics which impede gambling harm prevention; strong restrictions on gambling-related marketing; and an independent public health-based education programme. Additionally, Price et al. [8] argue that operationalising a public health approach to gambling harms also requires five strategies: health promotion; health protection; disease prevention and harm minimisation; population health assessment; and health surveillance.

As the above examples highlight, a PH approach requires a multifaceted response with a range of initiatives and interventions not only to treat GRH on presentation, but also to *prevent* them from occurring in the first instance. Recent reports have identified that the targets and strategies of PH approaches to GRH must: recognise that the input of those with lived experience is integral [10]; understand product-based risks [11–13]; and include targeted advocacy and campaigning [14, 15]. Other authors have suggested that initiatives should target affected others, wider communities [16] and entire populations [9, 17] whilst also targeting specific at-risk communities [18–20].

To minimise GRH most effectively, frameworks which identify problems and present potential mitigating strategies are necessary [7]. A range of frameworks already exist, but they vary in their content and intent. These frameworks are introduced in Table 1. Some frameworks have applied a pre-existing theoretical or methodological lens to describe gambling behaviour in relation to underpinning mechanisms or explanatory factors [21–25]. Several have focused predominantly on the conceptualization of harm and subsequent harm minimization strategies [17, 26–29] with others focusing almost exclusively on strategies to minimise harm [30, 31]. These harms-focused frameworks generally agree upon the types of harms experienced, even if demarcations between categories vary. Finally, several harms frameworks recognise the importance of targeting different sections of society [5, 9, 17, 26].

There are, however, two themes which appear consistently throughout the frameworks in Table 1: (1) a socio-ecological approach which recognises the need to focus on the relationship between harms and individual, community and societal determinants, and (2) the use of an established harms framework that is specific to the complex impacts of gambling [9]. Far less prominent are the range of goals and strategies that might make up a 'public health' approach to GRH. Thus, the construction of a comprehensive conceptual framework which can unify stakeholders towards GRH requires an exploration of the breadth of possible PH goals and strategies. Additionally, a comprehensive conceptual framework should be nuanced for the different strata of society which may experience GRH [7].

Our proposed conceptual model for GRH as a PH issue can help to overcome two significant barriers. First, the absence of a synthesizing framework makes mapping and evaluating the discrete aims of *cross-disciplinary* research or *applied settings* difficult. Secondly, no tool exists for organizations to evaluate current resource and service allocation. A PH approach requires rigorous and high-quality research evidence to inform decision-making [32–36]. Tools to support this understanding will help to identify opportunities for new organizations or future initiatives [37, 38]. A shared or common conceptual framework would facilitate knowledge transfer between key stakeholders within different disciplines. Therefore, a shared framework would support the mapping of key research and initiatives in the gambling landscape and make communication and coordination easier amongst a variety of stakeholders.

**Table 1. A summary of extant GRH frameworks categorised by topic.**

| Authors | Topic | Summary |
|---|---|---|
| Thorne et al. [42] | EGM | A consumer-focused framework for understanding how players interact with electronic gaming machines (EGMs) and the combinations of environmental features that motivate machine use. |
| Eby et al. [43] | Family and work | A strain-based framework to understand how gambling behaviour impacts work and nonwork engagement and performance. |
| Pattinson & Parke [21] | Theoretical framework applied to gambling | A grounded theory approach to gambling motivation on older British adults. It describes four important motivators: psychological stress reduction; physical stress mediation; stimulation; and accessibility. |
| Reber [44] | Game classification | A framework for classifying gambling activities based on the expected value of a game (negative vs positive) and the inherent flexibility (low vs high) of that game. |
| Korn & Shaffer [26] | Harms | A framework that categorises harm in relation to significant financial problems, family disruption, domestic violence, criminal behaviour, suicide and suicidal ideation, psychiatric conditions, gambling disorder, underage gambling and alcohol or substance use problems. |
| Wardle et al. [17] | Harms | A harms framework with three categories: resource, relationship or health. It also details sub-categories and indicators of harm. This framework also recognises the importance of stratifying harms by the social-ecological model. |
| Latvala et al. [27] | Harms | A framework which categorises harm into financial costs, labour costs, as well as health and wellbeing costs. In this framework, it is recognised that these costs have corresponding benefits. |
| Gainsbury et al. [28] | Harms | A behavioural science framework for stakeholders (individuals, community groups, industry, government, regulators, financial institutions, researchers) to reduce harm associated with online gambling. Specifically, it summarises potential facilitative strategies and actions. |
| Marionneau et al. [29] | Harms | A review of existing harms frameworks. It categorises harms as either related to finances, relationships, emotional and psychological, physical health, culture, work and study, or crime. |
| Langham et al. [5] | Harms | A conceptual framework and taxonomy of GRH. It categorises harms as either related to finances; relationship disruption, conflict or breakdown; emotional or psychological distress; decrements to health, cultural harm; reduced performance at work or study; and criminal activity. It conceptualises these harms as being experienced along a temporal continuum from general harm to legacy harms. It also proposes taxonomies of harm for individuals, affected others and communities. |
| Hilbrecht [9] (see also Abbott et al. [45, 46]) | Harms | A conceptual framework of factors which contribute to or influence harmful gambling, as well as how they interact. It includes general factors such as cultural, social, biological and psychological factors. It also includes gambling specific factors such as exposure, environment, resources, and types or gambling. |
| Browne et al. [47] | Measuring public health impacts | A framework combining indirect elicitation with a propensity score weighting. It outlines a method to describe the impact of risk factors on GRH likelihood. Then a causal model links GRH with specific health and well-being outcomes. |
| Costes [30] | Policy evaluation | A logical model describing what policy decisions may result in over the short and long-term. |
| Blaszczynski et al. [31] | Prevention strategies | A list of principles to guide stakeholders who wish to adopt and implement measures and prevention efforts that reduce GRH. |
| Dickson et al. [48] | Prevention strategies | This paper presents a risk behaviour model for adolescents, describing the relationships between risk and protective factors, risk behaviour and lifestyle, and health outcomes. |
| Afifi et al. [49] | Public Health Approach | A conceptual framework for studying problem gambling that relates family history, social/psychological variables and gambling exposure to gambling behaviour and problem gambling. It then relates problem gambling to health conditions, health and functioning, and help-seeking. |
| Lamont et al. [50] | Regulation | Proposes a conceptual framework to facilitate interdisciplinary research into corporate social responsibility of gambling operators in relation to sport sponsorship. |
| Delfabbro et al. [51] | Regulation | A collaborative, integrated and evidence-based framework for gambling product safety regulation that brings together key principles, stakeholders and approaches that encourage reform. |
| Lee [52, 53] | Relationships and treatment | Both articles outline a framework for understanding pathological gambling in a family-based therapeutic environment. |
| Zangeneh et al. [22] | Theoretical framework applied to gambling | Applies 'Problem Behaviour Theory' to adolescent gamblers, which outlines links between personality, environmental and behavioural factors that may determine gambling behaviour. |
| St-Pierre et al. [23] | Theoretical framework applied to gambling | Applies the 'Theory of Planned Behaviour' to adolescent gambling, relating attitudes, subjective norms, perceived control to intentions and future behaviours. |
| Mills et al. [24] | Theoretical framework applied to gambling | Applies Self-Determination Theory to gambling behaviour that highlights the role of motivation and need frustrations in developing problem gambling behaviour. |

*(Continued)*

**Table 1.** (Continued)

| Authors | Topic | Summary |
|---|---|---|
| Parke and Griffiths [25] | Theoretical framework applied to gambling | Applies grounded theory to the effects of technological developments on gambling behaviour. It proposes that information technology developments have elevated gambling involvement through increased outcome control, reduced discipline, expediency and consumer value. |
| Walker et al. [54] | Treatment outcome framework | Outlines a framework to be used by researchers when reporting the outcomes of problem gambling treatment. |
| Province of British Columbia [55, 56] | Strategy Document | Three goals of reducing incidence of problem gambling, reduce harmful impacts of excessive gambling and ensure gambling is delivered in a manner that encourages healthy choices. This should be achieved by public awareness and communication, education and training, risk management, treatment services, policy, research, industry training and information management. |
| Province of British Columbia [57, 58] | Strategy Document | The strategy has long-term objectives which are closely related to their three goals: to raise public awareness of gambling risks; encourage responsible gambling and informed choices; and to provide effective treatment and support for those affected. It also outlines eight approaches to support the goals: public awareness; education and training; develop responsible gambling strategies; develop policy; perform evaluative research; industry training; information management systems; via treatment services. |
| New Zealand Ministry of Health [59] | Strategy Document | Adopts Korn and Shaffer's [26] continuum of need and intervention. It also aligns their strategy with the He Korowai Oranga: Māori Health Strategy. Generally, the three interconnected tenets are healthy individuals, healthy families and healthy environments. These three elements feed into the overall aim of a healthy future. Underneath this relationship sit the various directions, key threads and pathways which describe strategies and other aims. |
| New Zealand Ministry of Health [60] | Strategy Document | Presents a framework for achieving strategic objectives. It has a general aim of reducing GRH and related health inequalities, with 11 objectives and 5 actions. |
| New Zealand Ministry of Health [61] | Strategy Document | Proposes increasing access to services that are responsive and targeted and developed with people with lived experience. Also proposes public health initiatives with the aim to increase awareness and engagement amongst those at risk, specifically to address stigma and education-based strategies. They also suggest enabling a gambling harm workforce, addressing cultural and language barriers, developing digital services and promoting action-oriented research to evaluate and improve services. |
| Gambling Commission [62] | Strategy Document | A four-stage public health approach to define the problem, identify risk and protective factors, develop and test prevention and treatment strategies, and ensure widespread adoption. They suggest this should be delivered through two vehicles of prevention and education, and treatment and support. |

Furthermore, a conceptual PH framework for GRH could benefit from lessons learned in other commercial, legal but potentially harmful, sectors, such as the alcohol, tobacco, and products high in fat, salt and sugar (HFSS) sectors. Although these comparisons are not widely prevalent within the frameworks highlighted in Table 1, previous research has explored how specific interventions or approaches within other sectors can inspire similar strategies towards GRH. Friend and Ladd [39] evaluate how a PH approach to GRH could learn from interventions to curb tobacco advertising. Thomas et al. [40] explore the opinions of PH experts within these industries and find that industry actors provide a barrier to the instigation of PH policies through political lobbying and donations, thus highlighting the need for a delineation between policymakers and industry. Other work highlights how industry actors use messages around the complexities involved with deploying a PH approach to deter such an approach being taken [41]. A successful PH approach to GRH—with inspiration from other sectors—should be informed by approaches in those sectors which are successful in reducing harms. Accordingly, the aim of this paper is to explore how PH approaches to GRH could learn from the tobacco, alcohol and HFSS sectors. We propose a *conceptual framework* which aligns disparate PH strategies and approaches—and potentially unites sector stakeholders—towards the prevention of gambling-related harms. Moving beyond the frameworks we have explored above, our framework was constructed first of all by carrying out a scoping review of extant PH approaches towards alcohol-, HFSS-, tobacco-, and gambling-related harms. This explored—

and developed a categorisation of—PH approaches found within each sector. Approaches to crime were also part of the original research focus but were excluded as we are concentrating only on legal products. We then categorised the PH approaches found during our scoping review and intersected them with the socio-ecological gambling-harms model proposed by Wardle et al. [17], allowing conceptual relationships to be drawn between potential PH strategies and goals, different levels of society, and different forms of GRH.

This paper is structured as follows. Firstly, we outline the methodology of the scoping review of previous PH approaches towards alcohol-, tobacco-, HFSS-, and gambling-related harms. Secondly, we evaluate the findings of the review, derived from a narrative analysis of the strategies prevalent within the sample of literature. Thirdly, the strategies and goals which emerged from our scoping review are developed into a conceptual framework for GRH. This also incorporates a socio-ecological model and a categorization of GRH outcomes by type of harm as well as severity and temporal experience of harm. Finally, we highlight how the framework can be used by a range of stakeholders towards the development of interventions which minimise GRH.

## Methods

We conducted a scoping review to explore existing PH approaches to GRH, and how they could learn from the tobacco, alcohol and HFSS sectors. Our initial focus also included crime-related harms, given their impact on society, communities and individuals alike. However, the decision was made during the initial search to concentrate only on legal products. This was because the significantly different regulatory and economic relationship that industries supporting crime have with society and government given their illegality meant this topic was considered the 'least' relevant. We followed PRISMA guidelines [63] for the identification, screening, eligibility and inclusion of papers, detailed in Fig 1. We preregistered the scoping review on the Open Science Framework (OSF) (https://osf.io/d7js2), and broadly followed the five stages as recommended by Arksey and O'Malley [64]. Our first stage consisted of the identification of the scoping review's guiding research question: what is a public health approach to GRH and how can it learn from other sectors? To answer this question, our aim was to identify existing PH approaches towards tobacco-related, HFSS-related, alcohol-related harms alongside those towards GRH. These approaches would then be synthesized to develop a categorisation of PH approaches to GRH that would be developed into a collaborative framework.

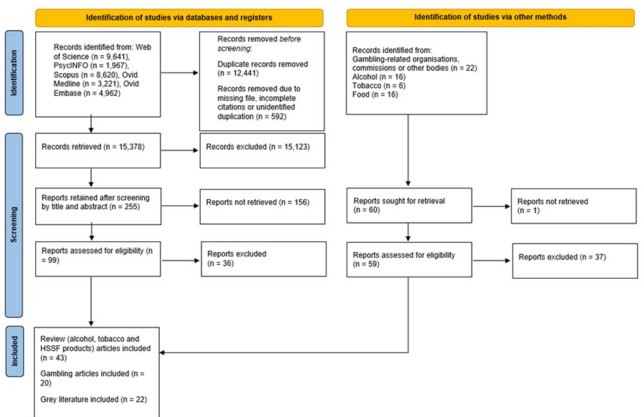

**Fig 1. PRISMA flow-chart of record identification and overall screening counts.**

The second stage was the identification of relevant studies. We (BF; JW) began this process with an initial search in November 2022, with search terms (public health) AND (approach OR framework OR tackling)) AND (gambl* OR tobacco OR alcohol OR crime OR fast food) entered into the Web of Science, PsycInfo, Scopus, Ovid Medline and Ovid Embase databases. We first selected 'fast food' as a search term but our initial search omitted research focused on other HFSS products such as sweet, salty and high fat food and beverages. We therefore conducted an additional search in January 2023 for papers related to HFSS products, using search terms ((public health) AND (approach OR framework OR tackling)) AND (food AND fast OR processed OR (high AND sugar OR salt OR fat) into the aforementioned databases. Eligible papers were those which adopted a specific PH focus towards the respective harms of each sector. The other inclusion criteria stipulated a focus on highly developed economies and articles written in English. With the research conducted within Great Britain, we sought papers published after 2005, the year in which the Gambling Act—which controls gambling in Great Britain—was given royal assent and which transformed gambling in Britain into a heavily advertised, deregulated commercial activity [1]. Once records had been identified from academic databases, we downloaded citation files and added them to Mendeley. We removed duplicates and then imported the remaining citation information into Excel. Additionally, we searched relevant websites for grey literature (see S1 Appendix), using the search terms "public health approach" or "public health framework" or "public health". Grey literature was sought from relevant organisations to provide insight beyond peer-reviewed journals. We downloaded relevant grey literature full texts and stored them in an online folder. Citation information was added into the Excel file by hand. We used the grey literature to compare approaches explored in peer-reviewed journals with those highlighted by non-academic organisations such as Alcohol Focus Scotland [65], or the Gambling-Related Harm All Party Parliamentary Group [66]. However, given the difference in robustness between peer-reviewed and grey literature, our findings are based solely upon those found within peer-reviewed literature.

Our initial searches returned a working sample of 15,378 titles after deduplication. We (JW and BF) then sifted through the working sample (N = 15,378) according to the inclusion criteria by title, abstract and full text. Our first sift by title saw the working sample reduced, according to the aforementioned criteria, to a new working sample of 1,037. At this point, we retained and separated alcohol-, HFSS- and tobacco-related review papers and gambling-related research papers into a new Excel sheet. We prioritised review papers for alcohol-, HFSS-, and tobacco-related harms (rather than papers describing individual studies) due to the short timeframe of the scoping review, and the inclusive nature of these reviews. Our second sift across both searches involved screening abstracts against the inclusion criteria and resulted in retaining 255 papers. Full details of excluded papers—and the reasons for exclusion—can be accessed through the OSF link. We (JW and BF) screened 231 alcohol-, tobacco- or HFSS- review papers by title, abstract and full-text, resulting in 43 retained for the fourth stage of data charting. One example of inclusion was Crombie et al.'s [67] review of interventions designed to prevent or reduce alcohol-related harms. Published after 2005, their review highlighted the range of interventions deployed in a range of advanced economies which can be adopted as a PH approach.

We reviewed individual gambling articles as per the original protocol, seeking empirical studies which either sought to explore the implementation of a public health strategy, or explored the requirements for such a strategy with findings drawn from participants' behaviours or against the background of wider socio-economic, or commercial determinants. We also retained conceptual articles which we felt provided context to the strategies found within empirical articles. We screened 24 gambling-focused articles by full-text (two additional articles were identified from full-text reading), resulting in 20 being retained for data charting. An example of a gambling-related paper that fulfilled the inclusion criteria was Kolandai-Matchett

et al.'s [33] study, which explored the implementation of a PH approach towards GRH in New Zealand.

Following Arksey and O'Malley [64], the final stage of our scoping review was the narrative analysis of themes prevalent within the sample of literature. We (JW and BF) analysed the specific PH-related approaches within each individual study or review paper. The full categories of data we extracted are introduced in Table 2. We focused on the data extracted under the category of 'Summary of Findings and Themes of Public Health'. Our narrative analysis grouped PH approaches by coding strategies and interventions according to type, thus developing a broader categorization of strategies which can be applied to reduction or prevention of GRH. We also coded the end goals or aims of strategies, thus resulting in broad PH goals which, when achieved, may result in the prevention or reduction GRH. We (JW and BF) coded strategies and goals separately, but found upon comparison that our respective analyses of goals and strategies returned similar results. After negotiation over the definition and categorization of goals and strategies, we ended with three broad PH goals which could be achieved by three broad PH strategies. We define these PH goals and strategies in the results of the narrative analysis section that follows below. These goals and strategies apply across tobacco, alcohol, HSSF and gambling.

**Table 2. Full categories for data extraction.**

| Category | Definition |
|---|---|
| Name | Name(s) of the author(s). |
| Year | Year of publication |
| Title | Title of the review or article |
| Link | The URL Link or DOI identifier for the study or review. |
| Declaration of Funder and Conflict of Interest | The source of funding for the study or review, and any declaration of conflicts of interest. |
| Country of Focus | Countries or jurisdiction under focus within each study or review. |
| Research Question | The research question(s) which guided the study or review. |
| Sample | The sample of participants within the gambling-related studies, or the number of papers explored within the alcohol-, HFSS-, and tobacco-related reviews. |
| Types of studies included (*Reviews from other sectors only*) | The types of studies included within reviews. Studies may include intervention-based studies and studies of policy documents. This field also included the search terms used by each review (where given) to identify relevant studies. |
| Design (*Studies in GRH only*) | The methodology used within gambling-related studies. |
| Intervention | The intervention(s) under focus, designed to prevent or reduce harms. |
| Outcome Measures | The measures upon which the interventions are designed to impact. For example, outcome measures within gambling-related studies were evaluated through participants' measurements on gambling screens such as Problem Gambling Severity Index or South Oaks Gambling Screen. |
| Summary of Findings and Themes of Public Health | A summary of findings within each study or review, with a focus on their implications for a public health approach. Data were abstracted for the narrative analysis of strategies deployed, as well as the main aim or outcome.<br>Summaries developed from tobacco-, HFSS-, and alcohol-related reviews were based on the sample of studies reviewed within each paper. Summaries from gambling-related papers were based on results from the approach which was applied, or upon recommendations for future strategies based on analysed data. |
| Limitations | Limitations of the study or review under analysis. This included limitations given by their author(s), in addition to those highlighted by the authors of the current scoping review. |

## Results of narrative analysis

The sample of alcohol-, HFSS-, and tobacco-focused reviews and gambling-focused studies (N = 63) is introduced within Table 3, alongside the interventions explored within each paper. Table 3 highlights the wide range of journals and jurisdictions present within the sample. Whilst the interventions and approaches varied, our analysis of data from the scoping review found that interventions were driven towards achieving three broad PH goals and three broad PH strategies. These are discussed in the following two sections.

### Three broad public health goals across potentially harmful product sectors

The PH Goals identified were (1) the prevention of harms, (2) the regulation of industry, and (3) support for those experiencing harms. This reflects the need to include both health promotion and an understanding of the epidemiology of non-communicable diseases. We discuss these three goals below.

We define the goal of prevention of harms as the need *to prevent harms from occurring at the earliest opportunity through societal level awareness and destigmatisation*. We highlighted prevention-focused strategies as occurring through health promotion campaigns which sought to denormalise harmful behaviours, whether through 'responsible drinking' campaigns [67, 68], the provision of healthier alternatives to HFSS products [69, 70] or the delivery of school-based awareness campaigns to reduce tobacco consumption [71, 72]. Also prevalent across the reviews into alcohol-, HFSS-, and tobacco-related harms, was evidence of mass-media or educational campaigns which generate societal-level awareness. Our analysis, however, highlighted a lack of evidence of strategies which seek to destigmatise GRH. There is a perceived need according to stakeholders to increase public awareness of GRH [33, 73], as well as to understand how different contexts may lead to GRH [74–76].

We define the goal of the regulation of industry as the need *to prevent harms through the central management of industries and their products*. How regulation should occur differed within the sample depending on the harm under study. Themes common across the alcohol-, HFSS-, and tobacco-focused reviews included the management of political lobbying [77], and the taxation of products [70, 78–85]. As with the goal of prevention of harms, studies into the prevention or reduction of GRH identified a need for stronger regulation of the gambling industry, as opposed to offering evidence of the efficacy of legislative or regulatory measures already in place. Studies of stakeholders within the industry highlight the need to manage industry involvement within political lobbying processes [74], and the need to regulate specific products which are perceived as harmful [74, 75]. Regulatory measures towards the prevention of GRH could therefore learn from approaches to harms within other sectors. Whilst the PH goal of regulation was prevalent across all sectors, only studies into GRH were unable to provide any evidence of the efficacy of regulatory measures.

We define the overarching goal of support of those experiencing harm as *the need to treat harms already experienced through targeted, specialist help*. This theme of targeted support was prevalent across every sector, although the strongest evidence linking targeted support to the reduction of harms was found within tobacco-related reviews [70, 82, 86, 87]. There was also evidence within the other sectors of targeted support incorporating family or affected others [88] or community involvement [89]. Data from our scoping review again found that evidence bases within the other sectors were more developed in relation to support, than within the gambling sector. In contrast to the other sectors under study, studies into GRH provided evidence from stakeholders on how fulfilling the goal of support could lead to the reduction of harms [33, 73, 74, 90, 91]. However, the evidence linking targeted support to GRH still

**Table 3. Sample of academic literature and identified public health goals and strategies.**

| Author | Jurisdiction | Journal/Publication | Intervention | PH Strategy | PH Goals |
|--------|-------------|---------------------|--------------|-------------|----------|
| Alcohol-Related Reviews (n = 11) | | | | | |
| Anderson et al. [108] | None specified | *Alcohol and Alcoholism* | Typology of drinking cultures and impact of community-based interventions on the wider community as opposed to the individual. Authors could not identify any attempts to influence PH through the change of social norms. | EA | Prevention |
| Calabria et al. [88] | None specified | *Addiction* | Identified interventions include: BMI Project CHAT (1 session); Medication; CBT and family therapy sessions (12–15 sessions); Family therapy (25–32 sessions); CBT-based peer group intervention (25–32 sessions); Community reinforcement approach (12 sessions). None of the studies reviewed indicated any strong methodological approaches. Most promising approaches included CBT, family therapy and community reinforcement. | SMI | Support |
| Cho and Cho [93] | None specified | *International Journal of Environmental Research and Public Health* | Educational programmes explored across the sample, aimed at altering the drinking behaviour of adolescents (13–18 years). Analysis demonstrated ability of programmes to reduce the *amount of alcohol consumed* by teenagers per session. | EA | Prevention |
| Crombie et al. [67] | Australia, Canada, Denmark, England, Ireland, Japan, New Zealand, Northern Ireland, Scotland, Sweden, USA, Wales | *Alcohol and Alcoholism* | Review explored wide range of interventions, including taxation and price controls, legislation and enforcement, drink-driving, marketing, drinking environment, high-risk groups, problem drinkers, education programmes. Interventions across countries under focus strongly aligned on education as well as assistance to those experiencing harm. | EA, UEP, SMI | Prevention, Regulation, Support |
| Dietrich et al. [92] | Focus on middle and high school age children | *Health Education* | Randomised controlled tests based either on universal programmes or a targeted approach towards high-risk teenagers. Findings highlight the importance of theory of behaviour change in the design of intervention, as well as applied segmentation to target support where needed. | EA, SMI | Prevention, Support |
| Drummond et al. [109] | European Union | *Addiction* | Review of the availability of alcohol interventions in Europe. Findings highlight gaps and future research questions around the characteristics of alcohol intervention systems in a range of European countries, the intervention service provision in Europe, and the gap between the prevalence of alcohol-usage disorder and availability of intervention in Europe. | SMI | Support |
| Gray et al. [68] | Various | *Health Communication* | Interventions which seek to promote 'responsible drinking', including media-based interventions (adverts and posters), education/Psycho-education, remote/e-health resources, community-based interventions (group). Twenty-one of the 55 studies reviewed demonstrated a favourable cognitive response to 'responsible drinking' messages. | EA, SMI | Prevention, Support |

*(Continued)*

**Table 3.** (Continued)

| Author | Jurisdiction | Journal/Publication | Intervention | PH Strategy | PH Goals |
|---|---|---|---|---|---|
| Martineau et al. [101] | None specified—focus on societal level interventions. | *Preventive Medicine* | Numerous interventions were identified across ten different policy areas: alcohol server setting, alcohol sales availability, illicit alcohol, taxation, mass media, drink-driving, schools, HE settings, family/community interventions, workplace. Key interventions emerging as most effective included: restricting hours or days of sale, reducing outlet density, minimum legal drinking ages, drink-driving checkpoints, increased police patrols, drink-driving awareness campaigns, mass media campaigns. | SMI, UEP | Prevention, Regulation, Support |
| O'Donnell et al. [110] | None specified | *Frontiers in Psychiatry* | Two brief interventions. Short, personalised feedback sessions, or more intensive intervention such as counselling, motivational interviewing, or CBT. Paper reports positive nature of brief interventions. | SMI | Support |
| Patra et al. [106] | None specified | *Contemporary Drug Problems* | Regulation of alcohol taxation and pricing at the point of sale. Almost all 54 studies reviewed found some impact as expected. An increase in alcohol tax or price per unit was associated with reduced high-risk drinking, and alcohol-related harm. | UEP | Regulation |
| Wilkinson et al. [105] | Australia, England and Wales. | *Public Health Research & Practice* | Restrictions on trading hours. Findings from Australia demonstrate a relationship between reduced hours of sale and substantially reduced rates of violence. Reduced impact of trading hour restrictions found in England and Wales. | UEP | Regulation |
| HFSS-Related Reviews (n = 16) | | | | | |
| Aranceta and Perez-Rodrigo [111] | Global | *British Journal of Nutrition* | Recommendations in relation to dietary intakes, nutritional goals, and guidelines for fat/fatty acids across different jurisdictions. Authors highlight the need for standardised protocol or intake recommendations to be applied globally. | EA | Prevention |
| Backholer et al. [80] | Australia | *Public Health Nutrition* | Tax on sugar-sweetened beverages. Tax on beverages is likely to lead to reduced weight and consumption across all socio-economic groups. However, tax would be most punitive for lower socio-economic status groups as they would pay a greater proportion of their income in additional tax. | UEP | Regulation |
| Capacci et al. [78] | Europe | *Nutrition Reviews* | Policy documents from Europe demonstrates specific evidence of a wide range of interventions, including marketing restrictions, awareness campaigns, access to healthier alternatives, and tax on unhealthy foods. | EA, UEP | Prevention, Regulation. |
| Fattore et al. [96] | Global | *Nutrition Reviews* | Various interventions reviewed across 36 studies. Review highlights the efficacy of educational programmes, dietary interventions, weight-loss interventions, and counselling. | EA, SMI | Prevention, Support |
| Gittelsohn et al. [112] | Various—Focus on community-based food settings. | *Preventing Chronic Disease* | Thirteen interventions explored based on social marketing and community influence. Cost-effective interventions (for example, clear labelling of foods as healthy) may have significant impact on prepared-food source sales and customer behaviour. | EA | Prevention, Regulation |

*(Continued)*

**Table 3.** (Continued)

| Author | Jurisdiction | Journal/Publication | Intervention | PH Strategy | PH Goals |
|---|---|---|---|---|---|
| Hession et al. [69] | UK | *Obesity Reviews* | Diet-based interventions, including diets with reduced carbohydrate content, 'healthy eating' advice, low fat diets, and calorie-deficit diets. Findings particularly highlight low carbohydrate/high protein diets as effective reducing cholesterol and blood pressure up to one year post-trial. | EA, SMI | Prevention, Support |
| Hyseni et al. [79] | UK | *PLOS One* | Interventions were categorised using nine stages along the agentic/structural continuum, from "downstream": dietary counselling (for individuals, worksites or communities), through media campaigns, nutrition labelling, voluntary and mandatory reformulation, to the most "upstream" regulatory and fiscal interventions, and comprehensive strategies involving multiple components. | SMI, UEP | Prevention, Regulation, Support |
| Kiszko et al. [94] | USA | *Journal of Community Health* | Display of calorie information of menus. Review found that most restaurant patrons are aware of calorie information on menus, but only certain groups (women, residents of affluent areas, and those motivated by nutrition information) are likely to be influenced by calorie information. | EA, UEP | Prevention, Regulation |
| Lhachimi et al. [113] | None specified | *Cochrane Database of Systematic Reviews* | Tax on saturated fat or total fat. No evidence found on the impact of tax of saturated fat or total fat. | None Identified | Regulation |
| O'Brien et al. [98] | UK and Australia | *Nutrients* | School-targeted, nutrition-based interventions. The review found that school-based approaches can have positive impact on the intake of fruit, vegetables, and fats. | EA | Prevention |
| Prowse [107] | Canada | *Health Promotion and Chronic Disease Prevention in Canada* | Possible interventions identified as part of findings which highlight the exposure of marketing to children in everyday settings. A comprehensive approach to restricting unhealthy food marketing to children that addresses product, promotion, place and price is required. | UEP | Regulation |
| Sinclair et al. [95] | USA | *Journal of the Academy of Nutrition and Dietetics* | Display of calorie information, in addition to extra context, on menus. Meta-analysis demonstrates that labelling menus with calories alone has no effect on calories selected or consumed. However, choice of calories was influenced by inclusion of contextual or interpretive information, such as exercise required per meal. | EA, UEP | Prevention, Regulation |
| Vezina-Im et al. [97] | Canada | *Public Health Nutrition* | Study evaluates the efficacy of school-based interventions towards the decreasing of sweet and sugared beverages. Legislative and environmental interventions identified as most effective, compared to educational or behavioural interventions. | EA, UEP | Prevention, Regulation |
| Von Philipsborn et al. [81] | UK and Germany | *Cochrane Database of Systematic Reviews* | Environmental interventions (for example: labelling, nutrition standards, economic tools, whole food supply, retail and food service, intersectoral, home-based) to reduce consumption of sweet and sugared beverages. Authors found moderate certainty that changes to labelling, economic tools (price controls), intersectoral collaboration, and home-based interventions can reduce consumption. | UEP | Prevention, Regulation |

*(Continued)*

**Table 3.** (Continued)

| Author | Jurisdiction | Journal/Publication | Intervention | PH Strategy | PH Goals |
|---|---|---|---|---|---|
| Wang and Bowman [114] | Global | *Current Atheroscleriosis Reports* | Population-wide interventions aimed towards the reduction of sodium intake. Comparison of approaches found six different economic analyses from six different countries. Evidence found generally supports population-wide interventions to achieve intake reduction, although evidence base from each country was small. | UEP | Prevention |
| Webster et al. [115] | Australia | *Cardiovascular Diagnosis and Therapy* | Interventions towards the reduction of salt intake. Interventions grouped by approaches: leadership and strategic approach, baseline assessments, implementation strategies (working with the food industry, changing consumer behaviour, labelling, interventions in public institute settings, advocacy), and monitoring and evaluation. Authors called for government-led approach in these areas, backed by monitoring and evaluation strategy. | EA, SMI, UEP | Prevention, Regulation, Support |
| Tobacco-Related Reviews (n = 14) | | | | | |
| Andrews et al. [89] | USA | *Nursing Clinics of North America* | Review of community-based participatory research approaches. Despite small evidence base, authors conclude that a well-designed community approach can help to reach marginalised communities. | SMI | Support |
| Bafunno et al. [99] | Global | *Journal of Thoracic Disease* | Review explored specific interventions: tax increases, public space bans, advertising and sponsorship bans, health warning labels, youth education, mass media campaigns. Authors report impact of tax increases, public space bans and mass media campaigns, although there is insufficient evidence of direct impact on consumption by marketing bans and health warning labels. | EA, UEP | Prevention, Regulation |
| Brown et al. [86] | None specified | *International Journal of Environmental Research and Public Health* | Parent smoking reduction and cessation interventions, anti-smoking socialisation interventions, targeted towards families with children aged 0–5 years. Interventions found to be more effective if focused solely on smoking cessation as opposed to additional outcomes. | SMI | Support |
| Burns et al. [84] | USA | *World Medical and Health Policy* | Various depending on state under analysis. Interventions linked with FCTC. Authors find that excise taxes show promise in reducing rates of tobacco use, as well as highlighting efficacy of smokefree and clean air policies. | SMI, UEP | Prevention, Regulation. |
| Chaloupka et al. [85] | None specified | *Tobacco Control* | Review of tax price controls. Authors find that majority of 18 papers reviewed present sufficient evidence that increased excise taxes and prices reduce tobacco consumption. Authors recommend that tax revenues should be used to fund control programmes and other health promotion activities. | UEP | Regulation |
| Ekpu and Brown [82] | Two sections for review: UK and international | *Tobacco Use Insights* | Various interventions explored and outlined as effective within the findings, including pharmaceutical or medical treatment, taxation, smokefree spaces, community-based interventions, mass media campaigns, school-based interventions, and employer-based interventions. | EA, SMI, UEP | Prevention, Regulation, Support |

*(Continued)*

**Table 3.** (Continued)

| Author | Jurisdiction | Journal/Publication | Intervention | PH Strategy | PH Goals |
|---|---|---|---|---|---|
| Forberger et al. [116] | USA and India | *Nicotine & Tobacco Research* | Scoping review of evaluations of smokeless tobacco control (SLT) policies. Authors find that SLT policies can result in increased awareness or reduced consumption. Positive effects also reported for health warnings, taxes, ban on flavoured products, and ban display of SLT. Higher education groups had higher awareness of campaigns and health warnings, whilst vice versa for lower education groups. | EA | Prevention, Regulation |
| Halas et al. [117] | Canada, United States, Europe, United Kingdom, Australia, and New Zealand | *Nicotine & Tobacco Research* | Review of extant research (658 reviews) to explore which tobacco control strategies have been previously explored. Authors find that intervention strategies linked to the FCTC. Seventy-one percent were related to a cessation strategy. 16 percent related to communication or education strategies, 7.6 percent related to protection from smoke exposure, 5.2 percent related to advertising and promotion, 2.9 percent related to pricing and tax controls, and 2.4 percent related to packaging and labelling. | EA, SMI, UEP | Prevention, Regulation, Support |
| Horn et al. [118] | Global | *Innovation and Entrepreneurship in Health* | Review of convergent innovation towards a PH approach towards tobacco control, according to social processes, organisational and institutional processes, financial processes, technological processes. Authors highlight importance of multisectoral cooperation, as well as community organisation and leadership. These can be aided by advanced financial and technological processes. | SMI | Prevention, Regulation |
| Hyndman et al. [87] | USA; Switzerland; Australia; Denmark; Germany | *International Journal of Environmental Research and Public Health* | Review of entry-level programme, curricular activity or component in smoking prevention or smoking cessation, and its impact on professional student practice in promoting client health. Review finds that brief tobacco counselling skills can positively impact patient quitting behaviours, although long-term impact is unclear. | SMI | Support |
| Linnansaari et al. [83] | Denmark, Finland, Iceland, Norway and Sweden | *Scandinavian Journal of Public Health* | Review of adoption and implementation of preventative tobacco policies in Nordic countries. Policies explored include taxation and price policies, protection from exposure to environmental tobacco smoke, product regulation, packaging, advertising and promotion, preventing product access by minors. Authors highlight efficacy of WHO directives on tobacco products, taxation, and advertising as beneficial for standards of prevention and control of tobacco. | EA, UEP | Prevention, Regulation |
| Mannocci et al. [72] | None specified | *Health Policy* | Intervention strategies which follow the FCTC: price and tax measures, non-price measures, and demand reduction measures. Despite lack of confidence in the efficacy of many interventions due to lack of evidence, authors highlight efficacy of price increases and combination of control approaches (including interventions in combination with smoke-free environments and media campaigns). | EA | Prevention, Regulation |

*(Continued)*

**Table 3.** (Continued)

| Author | Jurisdiction | Journal/Publication | Intervention | PH Strategy | PH Goals |
|---|---|---|---|---|---|
| Peirson et al. [71] | None specified | *Preventive Medicine* | Review of intervention studies on combustible cigarettes among children and youth in primary care settings. Despite small evidence base (9 studies), authors find that targeted behavioural interventions can stop children and young people trying and taking up smoking, as well as encouraging other young people to quit. | EA | Prevention, Support |
| Stead et al. [100] | Oceania, North America, Western Europe | *PLOS One* | Review exploring studies on consumer response to standardised tobacco packaging. Authors find that standardised packaging reduces attractiveness of tobacco as well as perceived quality. Standardised packaging also increased salience and effectiveness of health warnings. | EA | Prevention, Regulation |
| Multi-Sector Reviews (n = 2) | | | | | |
| Hoe et al. [77] | None specified | *Globalization and Health* | Scoping review of industry tactics to counter initiatives in tobacco, alcohol and sugar-sweetened beverage sectors. Authors found that industry tactics revolved around six key tactics: influencing policy, challenging unfavourable science, creating a positive image, manipulating markets, mounting legal challenges, and anticipating future scenarios. | SMI | Regulation |
| Mozaffarian et al. [70] | Global | *Circulation* | Systematic review of interventions designed to improve health behaviours in relation to the tobacco and HFSS sectors. Approaches grouped by: mass media and awareness, labelling, taxation, school-based interventions, local environmental contexts, and direct restrictions, healthcare system approaches, and surveillance systems. Strongest evidence highlights efficacy of taxation, school and workplace approaches, in addition to restrictions of marketing. Healthcare system approaches require training, systematic record keeping, in addition to mechanisms for feedback and project evaluation. | EA, SMI, UEP | Prevention, Regulation, Support |
| Gambling-Related Papers (n = 20) | | | | | |
| Adams et al. [119] | New Zealand | *Addiction* | Conceptual approach, asking what a PHA might look like based on the standard PH models and the New Zealand example. The New Zealand-based PH approach to GRH encompasses three strands: harm minimisation, health promotion, and political determinants, the latter of which is fulfilled through greater awareness around gambling-related harms and the need for transparency of industry involvement, surveillance mechanisms, and structural accountability. | Conceptual | Prevention, Regulation, Support |
| Afifi et al. [49] | Canada | *Canadian Journal of Community Mental Health* | Conceptual approach, seeking to develop a population health framework for gambling researchers. Framework towards a PH approach to gambling harm should centre around the awareness of genetic factors, physical environment and exposure, and social contexts. | Conceptual | Prevention, Regulation, Support |

*(Continued)*

**Table 3.** (Continued)

| Author | Jurisdiction | Journal/Publication | Intervention | PH Strategy | PH Goals |
|---|---|---|---|---|---|
| David et al. [34] | Australia | *Addiction Research & Theory* | Study of 50 stakeholder interviews exploring the challenges and opportunities for PH advocacy towards GRH. Findings highlight need to counter industry influences, as well as an adoption of frameworks similar to FCTC. The authors also suggest that a PH approach should include developing and enabling advocates, challenging the structural barriers created by industry, and developing consistent advocacy responses | EA, SMI | Prevention, Regulation |
| Delfabbro and King [120] | Australia | *International Journal of Mental Health and Addiction* | Authors explore why PH approach has not yet become the most prevalent approach to GRH. They highlight the need for a population-wide strategy, with targeted support in light of lack of existing evidence. Authors also highlight focus on destigmatization. | Conceptual | Prevention |
| Dickson et al. [102] | Switzerland | *International Journal of Environmental Research and Public Health* | Conceptual approach towards the effective monitoring of Swiss gambling industry. Three pillars identified: management of gambling industry, awareness campaigns on harmful gambling behaviour, healthcare and support | Conceptual | Prevention, Regulation, Support |
| Dymond et al. [121] | Wales | *The Lancet* | Correspondence highlighting the need for gambling-harm treatment clinics in Wales. Authors also highlight the vulnerability of the most deprived areas of Wales to GRH. | Conceptual | Support |
| Guilcher et al. [73] | Canada | *Health and Social Care in the Community* | Informed by feedback of 30 healthcare and service providers, authors explore how screening for problematic gambling behaviours can be more accessible to the local community. Multiple interventions outlined by service providers, grouped according to top level policies, screening tool, staff skills, screening procedures, team resources and support. Authors also highlight the need for increased public awareness of problem gambling. | EA, SMI | Prevention, Support |
| Howat et al. [32] | Australia | *Australian Journal of Primary Health* | Case study on PH advocacy to counter industry tactics. Key conclusions include the importance of research-led campaigning, the ability to challenge vested interests, intensive and prolonged media activity, collaboration, and the ability to be economical and pragmatic. | Conceptual | Prevention, Regulation |
| John et al. [76] | Wales | *Frontiers in Public Health* | Explores the current trends and patterns in gambling behaviour, as well as the salient social, cultural, and environmental factors in the development of harmful gambling behaviours. Study also explores the efficacy of support services. Findings highlight need to adopt an approach to gambling-related harms based on WHO guidelines: understand the nature of the problem, identify causes, evidence-based approach, and implement solutions. Authors also highlight the need to understand social, environmental and cultural factors which may be specific to each gambler. Density of gambling operators was also presented as a contributory factor, with Fixed Odds Betting Terminals highlighted as a specifically harmful product. | UEP | Prevention, Regulation |

(*Continued*)

**Table 3.** (Continued)

| Author | Jurisdiction | Journal/Publication | Intervention | PH Strategy | PH Goals |
|---|---|---|---|---|---|
| Johnstone and Regan [35] | UK | *Public Health* | Authors describe an approach to gambling as a PH issue which requires systemwide, multisectoral co-operation at local and regional levels. This approach is multi-faceted and includes: policymakers using the best evidence to prevent harm, understanding the prevalence of problematic gambling behaviour and its impact on affected others, ensuring that tackling GRH is a key PH commitment at levels of society, understanding how different assets or bodies can assist in prevent GRH, raising awareness sharing data, and involving those who have been harmed, ensuring all local authorities tackle GRH under a whole council approach, and the development of a whole-systems approach towards poverty and health inequalities. | Conceptual | Prevention, Regulation, Support |
| Kolandai-Matchett et al. [33] | New Zealand | *Harm Reduction Journal* | Study evaluating the implementation of a PH approach to gambling in New Zealand. The findings highlight how workplace and organisational policies towards GRH were developed, whilst positive influence on EGM-related policies were achieved, yet there was little influence on non-gambling fundraising policy. A PH approach was hindered by the ambivalence towards the seriousness of GRH, whilst collaboration with stakeholder organisations was also highlighted as important. Useful approaches included linking gambling to other comorbidities, highlighting best practice from other jurisdictions and evidence-based lobbying, as well as increasing public participation. | EA, SMI | Prevention, Regulation, Support |
| Kraus et al. [104] | Finland, Germany, Italy, Sweden, Norway, Australia, US | *Frontiers in Psychiatry* | Comparison of self-exclusion registers across jurisdiction. Self-exclusion registers were analysed as more effective when regulated centrally, compared to when operated by industry. | SMI | Regulation |
| Lischer et al. [122] | Switzerland | *International Journal of Environmental Research and Public Health* | Highlights the cross-over between gambling and loot-boxes. Viewpoint argues that the regulation of loot boxes is necessary from a public health perspective to protect children. | Conceptual | Regulation |
| Maltzahn et al. [74] | Australia | *Public Health* | Study of how GRH can be experienced by land-based bingo players. Findings call for five different mechanisms: Regulation of bingo product, delinking of bingo from digital products, remove political protection of industry, tailor interventions for subpopulation, address external factors responsible for gambling harm. | SMI, UEP | Prevention, Regulation, Support |
| Marshall [123] | Australia | *Journal of Gambling Issues* | Author highlights the role of local context in a PH approach, whilst also highlighting that gambling is different from other PH concerns due to its lack of biomedical problem, the lack of detrimental impact on most consumers, and the complex harms for the minority who experience problems. Author highlights importance of factors related to context and local community in the emergence of GRH. | Conceptual | Prevention |

*(Continued)*

**Table 3.** (Continued)

| Author | Jurisdiction | Journal/Publication | Intervention | PH Strategy | PH Goals |
|---|---|---|---|---|---|
| McCarthy et al. [75] | Australia | *Addiction Research & Theory* | Study of socio-cultural, environmental and commercial determinants which initiate or sustain EGM playing for women. The study proposes: Safe spaces (ensuring that other spaces/rooms away from EGMs are available for women who are alone), removal of incentives through refreshments, ensuring evidence-based information is available to gamblers, clear distinction between venue staff and harm specialists. | EA, UEP | Prevention, Regulation |
| Nguyen et al. [90] | Australia | *Fairfield City Health Alliance.* | Study which develops, implements and evaluates a gambling harm screening and referral system for general practice and community services. Five recommendations are made as a result of the findings of the project: a similar model should be rolled out to other general practitioners in the local area; community services are uniquely placed to implement the screening model; the system could be integrated with existing alcohol or drug screenings; the provision of an indicative screening outcome based on patient responses to guide interventions; the systematic gathering and storage of screening results to guide community-level initiative. | SMI | Support |
| O'Mullan et al. [91] | Australia | *BMC Public Health* | Study exploring how health and social service agencies can be improved to address needs of women seeking support for gambling-related intimate partner violence. Participants highlight the efficacy of timely referrals to other agencies within and outside the health sector. | SMI | Support |
| Shaffer et al. [36] | Global | *International Journal of Mental Health and Addiction* | Conceptual approach, integrating Reno and PH approaches towards the mitigation of gambling-related harms. Authors make four key conclusions: importance of research-led implementation of a PH approach, a PH approach derived from population-based observations, the need for PH initiatives which emphasise prevention and harm reduction, and a PH perspective which is balanced. | Conceptual | Prevention |
| Van Schalkwyk et al. [103] | UK | *The Lancet Public Health* | Viewpoint advancing the debate on the involvement of the gambling industry within policy formulation. The authors also highlight the need for a policy shift away from the commercialisation and profitability from gambling, towards one which recognises gambling as a PH issue. | Conceptual | Prevention, Regulation |

highlights the need to upscale support-led initiatives towards the reduction of societal-level harms [91].

## Three broad public health strategies across potentially harmful product sectors

In addition to delineating three common public health goals across harmful product sectors, our narrative analysis of scoping review data also identified three broad types of public health strategy that can be used to achieve these goals: (1) education and awareness (EA), (2)

screening, measurement and intervention (SMI) and (3) understanding environment and product (UEP).

The strategy of 'education and awareness' (EA) comprises initiatives which seek *to prevent or reduce harms through the provision of research-led information*. Within the alcohol sector, EA initiatives were aimed at children and adolescents [92, 93], whilst also seeking to encourage 'responsible drinking' [67, 68]. Within HFSS-related reviews, EA strategies were found within the provision of calorie or nutrition information [94, 95], access to healthier alternatives [69], and the use of widespread awareness campaigns [78, 96–98]. Widespread awareness campaigns on the harms of tobacco consumption were analysed as effective [71, 72, 82, 99], particularly when used to encourage changes in harmful behaviours at a young age [100]. EA strategies identified within gambling-related studies highlighted a need for initiatives and evidence towards the destigmatisation of GRH through their recognition as a PH issue [33], and increased awareness of GRH amongst healthcare and service providers [73]. Research also concurred that EA-led, mass awareness should be disseminated free from commercial discourse such as 'responsible gambling' which enables the industry's avoidance of responsibility and promotes the continuation of gambling regardless of level of harms experienced [75].

Secondly, Screening, Measurement, and Intervention (SMI) strategies are concerned with *the screening of harms*, *the subsequent intervention where required*, *and the measurement required to track the prevalence of harms*. SMI strategies—specifically intervention strategies—also comprise the management of industry practices, and their involvement within policymaking. Within alcohol-related reviews, interventions were geared towards the prevention of drink-driving [67, 101], as well as support for those who are already experiencing alcohol-related harms [67, 88, 92]. Within HFSS-focused reviews, SMI strategies consisted of the management of industry involvement within policymaking [77] and targeted dietary interventions [96]. SMI strategies explored within tobacco-focused reviews included targeted, cessation interventions [82, 86, 88], advice given to patients and the training of staff [70, 87], and the management of industry practice [77]. SMI strategies were also explored within tobacco-focused reviews through the adoption of the World Health Organisation's Framework Convention on Tobacco Control (FCTC) [84]. Within gambling-related studies, SMI strategies were again found, to provide targeted support to patients—and affected others [91]—through the *need* for more widespread support services [33, 73] whilst also being linked with treatment for other comorbidities. Targeted, specialist support would also be aided by the deployment of an easily accessible screening tool [73, 90]. The evidence shows strong support for the clear separation of the gambling industry from the government, as well as the transparency of lobbying by the industry itself [74]; and conceptual papers agree that regulation and policy should be immune to industry influence [32, 102, 103]. The separation of government and industry influence could also include the deployment of self-exclusion schemes which allow individuals to exclude themselves from gambling to avoid harms. Evidence presented by Kraus et al. [104] suggests that the state and the industry alike are ineffective at enforcing self-exclusion registers which can be easily circumvented. On the other hand, the authors still conclude that state-regulated registers are more likely to be effective at maintaining self-exclusion compared to those maintained by operators whose primary focus is revenue over harm limitation.

Strategies relating to the understanding of environment and product (UEP) *explore how harmful behaviours emerge within specific contexts or from specific products*, *in order to prevent or reduce further harms*. UEP strategies therefore seek to understand the social and environmental contexts which may lead to GRH and how these might be addressed. Alcohol-focused reviews provided three specific UEP strategies, namely reduced hours of sale [67, 105], reduced outlet density [67], and the adoption of tax and price controls to deter harmful consumption [106]. HFSS-focused reviews also explored the effect of taxes on the consumption of unhealthy

products [79, 80], the effective protection of children from marketing [70, 78, 96, 97, 107], the display of calorie information on menus [94, 95], and the impact of traffic light system labelling on the consumption of sugar-sweetened beverages [81]. Tobacco-focused reviews highlighted the efficacy of tax and price controls alongside the deployment of smoke-free spaces which denormalises smoking [82, 99]. As with the other two categories of PH strategy, gambling-focused studies highlighted a *need* for further UEP strategies to reduce harm, including more effective regulation of specific products—such as EGMs which were highlighted within the sample of studies as more harmful—which may encourage initial and continued harmful gambling behaviours [74, 75], as well as the social cues which may encourage a pathway to harmful gambling behaviours [76]. Indeed, interactions with specific products may also be intertwined with various social interactions with friends [76], or with staff [74], and a deeper understanding of how GRH may arise from specific environments would allow best practice (from stakeholders) or regulation to protect those at risk.

## Conceptual papers

Aside from the PH goals and strategies highlighted above, our scoping review also uncovered peer-reviewed papers whose approach was *conceptual* in nature. Conceptual approaches were mainly prevalent within papers whose focus was on the reduction or prevention of GRH. For the purposes of this paper, our grouping of conceptual papers included those which developed frameworks [36, 49], highlighted the necessary collaborations for a PH approach [35], or viewpoints which underlined the *need* for clear delineation between industry and policymaking bodies [103]. These papers were retained within the initial sample thanks to their valuable insight.

## Identifying existing frameworks

The categorisation of PH approaches that emerged from our scoping review was then used to develop a collaborative framework to address GRH. Our scoping review sought to identify literature that made recommendations for PH approaches or acknowledged wider socio-economic determinants of health in the gambling, tobacco, alcohol, and HFSS sectors. In many cases, the impact of PH interventions was not targeted at specific harms and approaches were intentionally operationalised to have general far-reaching impact. However, we recognised following our review that a framework that relates specific GRH to approaches requires categories of harms to be discernible. We also aimed to ensure that PH strategies identified from other areas were made relevant to gambling. It was thus necessary to include frameworks which have conceptualised GRH as this was not apparent in the articles identified from the scoping review. Therefore, separately from the scoping review but using the same databases, we identified a number of existing gambling-harm frameworks using the terms "gambl* AND harm AND (framework OR concept* OR strategy)". We then evaluated the commonalities and differences and made a pragmatic decision about which conceptualisation to incorporate into our framework.

The PH goals and strategies outlined above constitute the core of our conceptual framework. However, such strategies and goals often only target specific levels of society [9, 17, 27, 29, 124, 125] and individual categories of GRH [5, 26]. There are also considerable differences between gambling and the other sectors reviewed. Approaches to alcohol-, HFSS-, and tobacco-related harms generally focused on the amendment of products such as the reduction of fat content in HFSS foods, the regulation of marketing, education to encourage behaviour change, and taxation. GRH-related papers, on the other hand, highlighted the need to counter industry interests, the need for more effective awareness of—and screening for—GRH within

healthcare systems, and developing societal-level messaging that destigmatises GRH. Additionally, digitalisation affects gambling activities in a way that does not apply to alcohol, tobacco or HFSS consumption [1, 3]. Therefore, we do not wish to suggest that the PH approaches evident in the scoping review represent an exhaustive list, nor that all approaches in other sectors are necessarily effective for, or relevant to, gambling. Rather, by grouping the existing approaches into broad overarching categories, we hope the conceptual framework is able to categorise the variety of approaches currently in-focus, as well as those yet to be proposed. Furthermore, for those wanting specific strategy proposals, researchers have made attempts to list numerous potential strategies to tackle GRH from a PH approach [7, 8]. Our proposed conceptual framework is different because it offers a way to incorporate, and systematically organise, multiple PH goals and strategies while also taking into account the different social strata described in socio-ecological models and focusing on multiple categories of GRH.

In this section, we describe the process we followed to produce our conceptual framework. Firstly, we incorporate the socio-ecological model with the PH goals and strategies identified from the scoping review. Secondly, we then incorporate different GRH-related outcome categories, whilst also developing the framework to measure the severity and timescale over which GRH may be experienced. Following this process ensured that our framework was fully developed from the identification of PH goals and strategies, towards their application at different levels of society, and the prevention of GRH which may be wide-ranging and complex in nature.

## Incorporating the socio-ecological model of gambling

Previous discussions of PH approaches highlight the necessity for a framework to understand gambling impacts at various levels of society [9, 17, 27, 29, 124, 125]. The socio-ecological model is appropriate given its links to other PH domains [125–128] and its acknowledgement of the social and environmental determinants encountered by individuals which may also lead to GRH or shape people's experiences of them [125]. The socio-ecological model emphasises that situational factors interact with an individual's biopsychological characteristics [17]. For a long time, the focus of GRH research has been an individualised approach. We agree with others [7, 8] who argue that focus should be shifted to wider determinants of GRH. Our model draws on similar strata highlighted by Wardle et al. [17], moving from 'Individual' to 'Interpersonal' to 'Community' and 'Societal', as introduced in Fig 2. This stratification allows the targeting of intervention and treatment at specific parts of the population [9, 17, 29], and the understanding and mapping of the relations between various approaches or outcome goals. Public Health England (PHE) [129] used the socio-ecological model for a gap analysis of gambling risk factors, thus demonstrating the utility of the model as a tool to map research. For gambling specifically, the model shifts the focus from individual treatment approaches and explicitly recognises the harm caused to affected others [91], social groups [76], and at a societal level [9].

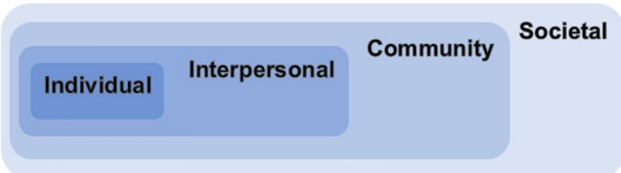

**Fig 2. The various nested strata of the socio-ecological model.**

The 'Individual' level focuses on biopsychological characteristics that might be classified as 'risk-factors' as well as issues such as individualised interventions. The 'Interpersonal' level focuses attention on the social and family structures of at-risk individuals, such as the partners of pathological gamblers or interpersonal interventions. The 'Community' level relates to local or online environments, community groups, alongside institutions such as schools, banks and workplaces. Crucially, the 'Community' level targets 'not-at-risk' groups as well as identified 'at-risk' groups. Finally, the 'Societal' level encompasses whole-population approaches such as public policy decisions, national campaigns, and macro-structures that determine legal and cultural practices [9, 17].

## Connecting the conceptual framework to different categories, severity, and timeframes of harm

Whilst developed against a socio-economic model to explore impacts against individuals, communities and societies, our framework still requires development in its relationship with specific harms. We therefore firstly developed the framework in accordance with categories of harm outcomes that are already well-established within gambling-related research. Secondly, we addressed how specific GRH can be experienced more or less intensely over different periods of time.

**Different categories of harm.** There are various conceptualizations of GRH. For example, PHE [129] classifies harms into financial; relationship; mental and physical health; employment and education; crime and anti-social behaviour; and cultural, with similar categories seen in other works [5, 29]. The discrete categories proposed by Langham et al. [5], PHE [129], and Marionneau et al. [29] are useful if GRH is the focus of a framework, although even they rely on further sub-categorisation to encompass specific outcomes. However, in seeking to develop an overarching conceptual framework with a considerable number of intersecting goals, strategies and approaches, we decided to adopt Wardle et al.'s [17] concise approach which categorises harms into three broad categories: *resources*, *relationships*, and *health*, underneath which sit six sub-categories and fifty indicators. Table 4 maps the more detailed GRH categorisations against Wardle et al.'s [17] simplified three-category framework. Crime is defined by Wardle et al. [17] as a resource-based GRH, as emphasis is placed on measuring the impact of gambling behaviour on organisations, systems and victims through the medium of money or resource cost. However, whilst we agree that this is useful for the measurement of harm and for public health decision making [130], it is important to note that the relationship between gambling and crime is itself multifaceted and complex [131] and, as it is not a legal activity, it was excluded as an area of focus from the scoping review. Crime that results from gambling, however, is still categorised as a GRH. Crimes that result from gambling may consist of anti-social behaviour, entail gambling as a contributory factor [17], or systemic crimes where regulations are not followed by operators (for example, non-compliance in relation to unfair practices, underage gambling, or unfair advertising) [131]. All categories of harm—resources, relationships, and health—may arise as outcomes from gambling behaviour, as well as forming determinants which influence gambling behaviour itself.

**Different severities and timescales.** Existing harms frameworks acknowledge that GRHs exist on a temporal continuum [5, 27] from brief (or *episodic)* to long-term, or even intergenerational. Additionally, the harms experienced from gambling at any level of society can range from inconsequential, to general or crisis-level [5]. As before, we have adapted Wardle et al.'s [17] categorization of harms within our conceptual framework for measurement across this continuum, introduced in Fig 3. These dimensions are a vitally important consideration when evaluating the implications or impact of gambling behaviour on individuals, social groups, communities, and societies in order to inform decision-making.

**Table 4. Synthesis of existing gambling-related harm frameworks according to Wardle et al.'s framework headings.**

| | Categories of Gambling-Related Harms [17]. | | |
| --- | --- | --- | --- |
| | Resources | Relationships | Health |
| **Korn and Shaffer** [26] | Significant financial problems<br>Criminal behaviour | Family dysfunction and domestic violence | Gambling disorder<br>Youth and underage gambling<br>Alcohol and other drug problems<br>Psychiatric conditions<br>Suicide, suicide ideation and suicide attempts |
| **Langham et al.** [5] | Financial harms<br>Reduced performance at work/study<br>Criminal activity | Relationship disruption<br>Cultural harm | Emotional or psychological distress<br>Decrements to health |
| **Latvala et al.** [27] **Costs and *Benefits*** | Financial costs<br>Labour costs<br>*Financial benefits (e.g., economic growth, revenues, infrastructure value)*<br>*Labour benefits (e.g., jobs)* | Health and wellbeing costs (e.g., relationship problems, socio-cultural costs)<br>*Health and wellbeing benefits (e.g., socio-cultural benefits through enhanced cultural identity and connectedness)* | Health and wellbeing costs (e.g., emotional stress, risk-taking behaviours, physical symptoms, suicidal thoughts)<br>*Health and wellbeing benefits (e.g., social activity, enhanced self-concept* |
| **Public Health England** [129] | Financial<br>Employment and educational harms<br>Crime and anti-social behaviour | Relationship disruption, conflict or breakdown<br>Cultural harms | Mental and physical health harms |
| **Marionneau et al.** [29] | Financial<br>Work and Study<br>Crime | Relationships<br>Cultural | Emotional and Psychological<br>Physical health |
| **Wardle et al.** [17] | Unstable employment<br>Job loss<br>Reduced performance<br>Debt<br>Financial insecurity<br>Reduced disposable income<br>Anti-social behaviour<br>Crime | Ruptured relationships<br>Neglected relationships<br>Exploited relationships<br>Reduced community cohesion/ participation<br>Social isolation<br>Increased inequalities | Reduced physical health<br>Psychological distress<br>Reduced mental health |

## Discussion

PH approaches encompass a range of goals and strategies, allowing for an understanding of complexity and nuance of societal health issues whilst retaining synergism. The core PH goals and strategies which emerged from our scoping review form the central interacting strands of a comprehensive conceptual framework for the prevention of GRH. This conceptual framework could also be helpful given that our analysis has demonstrated that the evidence base of a PH approach to GRH may be behind the curve compared to other sectors. To cement the shift away from focusing on individuals and reflect that public health approaches aim to minimise harm across families, social groups, communities and whole populations, our conceptual framework nests the PH goals and strategies within the socio-ecological model. To make it comprehensive, the conceptual framework also incorporates different categories of GRH as proposed by Wardle et al. [17], along with different severities and time scales.

Our full conceptual framework is introduced in Fig 4. By mapping how PH goals and strategies intersect with gambling harms and how the intersections can be stratified by the socio-ecological model, we propose a highly interactive framework that can be used in research, policy and practice. The three broad PH strategies intersect with the three PH goals. At each intersection, the model recognises their relationship to gambling outcomes: either resource-based, relationship-based, or health-based harms. Importantly, each strategy-goal intersection and related GRH can be differentiated by level of socio-ecological model, spanning from *individual*

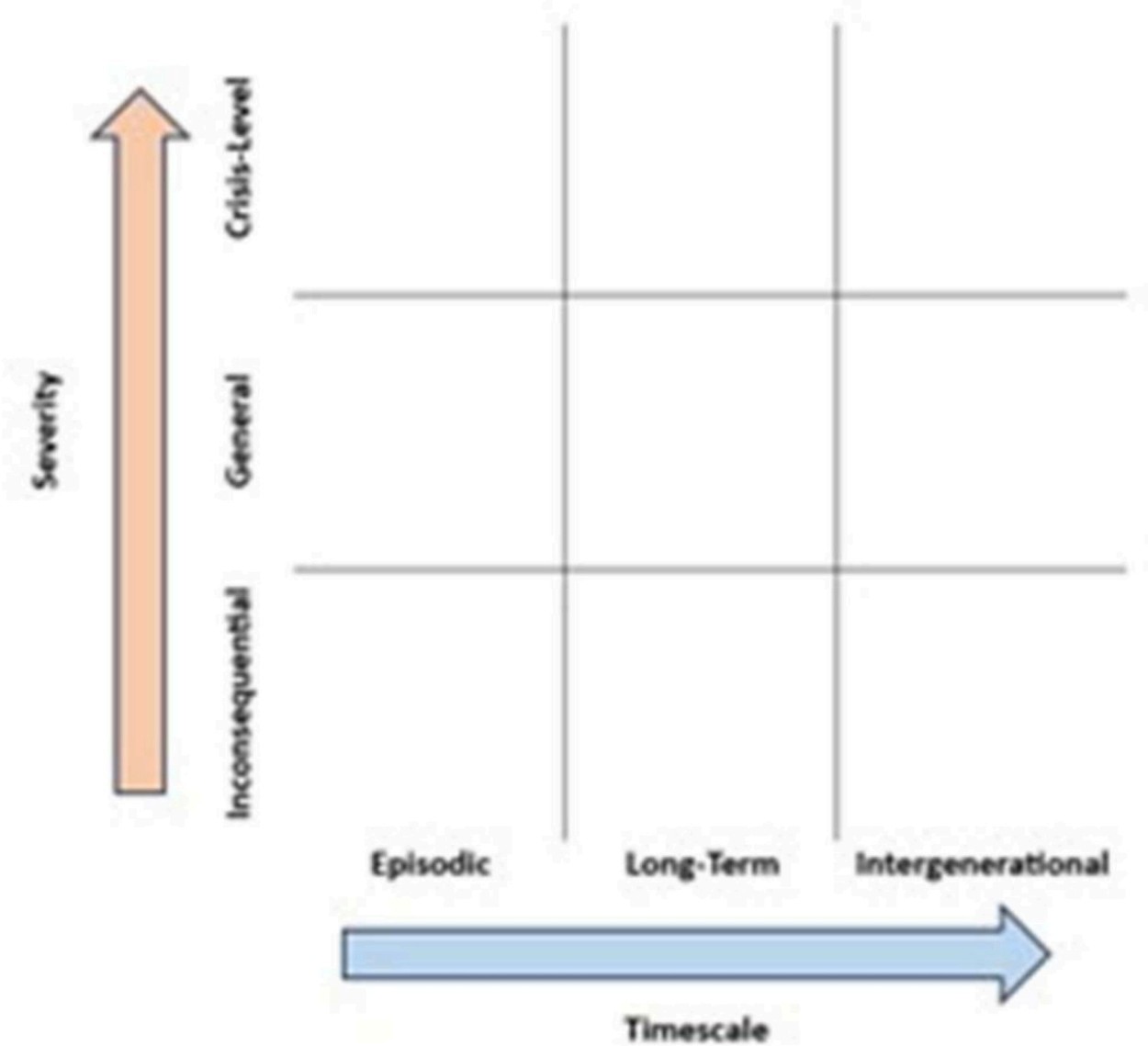

**Fig 3. An image depicting the important considerations of the temporal nature and degree of harms.**

to *family and social networks*, *community*, and *society*. Whilst many reports focus on 'selective' or 'at-risk groups' in the community [9], our framework additionally focuses on 'untargeted' approaches, given the importance of preventative measures in public health approaches at a societal level.

At a conceptual level, the tool allows researchers from varied disciplines to understand how their work intersects with society, specific GRH, and the wider gambling field. The framework will support the understanding of project related implications and impact. Furthermore, for stakeholders wishing to understand how a PH approach to gambling can be delivered or the types of considerations that need to be addressed, this conceptual mapping should prove useful. However, the framework's utility is most evident through its use as a tool for applied and research settings. The framework can be used for organising, evaluating, and strategising in a

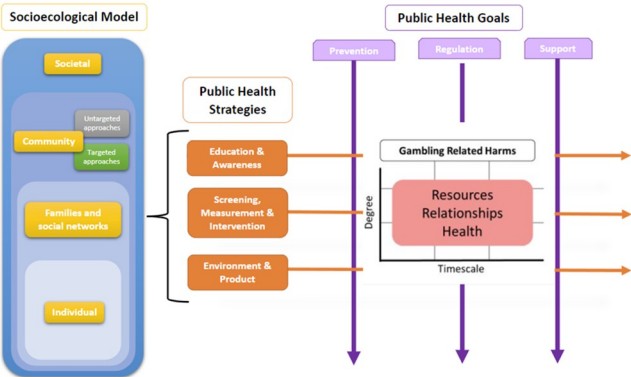

**Fig 4. A visual depiction of the full conceptual framework for the prevention of GRH.**

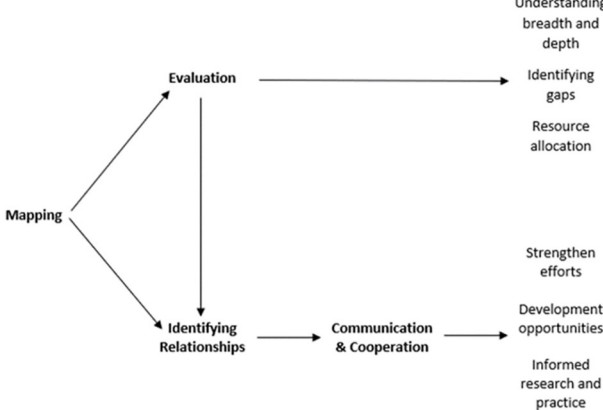

**Fig 5. A flow diagram to understand how the functions of the framework relate and lead to potential impact.**

research or service setting, with the subsequent benefit of facilitating communication and coordinated effort.

By populating the various intersections of the framework, stakeholders can systematically map research, policies, or services. The framework therefore serves four important functions, introduced in Fig 5. Firstly, the framework facilitates *mapping* of the breadth and depth of initiatives designed to prevent or reduce GRH, and, in doing so, identifies gaps in research or provision. Secondly, the framework enables the *evaluation* of interventions and approaches in relation to different levels of society or against the different PH goals or strategies. Thirdly, the framework enables stakeholders to *identify relationships* between different intersections of the framework, service provisions, research areas within or across disciplines, or the aims of research and applied settings. Finally, the framework facilitates the understanding relationships and establishes potential avenues for *communication and cooperation* across different stakeholder groups or disciplines. These functions are introduced in further detail below.

## The conceptual framework as a mapping tool

The primary function of the framework is that of *mapping*. Stakeholders could map support services at the intersection of SMI strategies towards the overarching goal of support at various

levels of the socio-ecological model, in relation to specific GRH. Stakeholders could use the framework to understand how provision varies across society. For example, the framework may highlight a lack of services focused at the 'families and social network' level, suggesting a need for greater provision for affected others. Additionally, stakeholders may find that services are focusing predominantly on 'health'-related harms such as psychological distress whilst the mapping highlights that greater focus should be placed on reducing financial harms. In this sense, interacting with the framework can provide information to support informed resource allocation and decision-making. Mapping could be done using a matrix, as demonstrated in Table 5 below, which demonstrates the interaction between socio-ecological model, PH goals and strategies, as well as gambling-related outcomes, including 'non-specific' outcomes identified by stakeholders outside of the broad categorization offered by Wardle et al. [17].

## The conceptual framework as an evaluation tool

A further, subsequent function for the conceptual framework is that of *evaluation*. Researchers could use the framework in a similar way to evaluate published literature. For example, a researcher may wish to evaluate the strength of evidence for awareness campaigns focused on safer gambling practices at different levels of society. This evaluation could focus on the intersections of EA strategies and the goal of prevention, as well as synthesise and evaluate the strength of evidence for other targeted, community, and whole population campaigns. This could then be easily translated into future campaign research or resource allocation in applied settings.

## The conceptual framework as a means to identify relationships

The third function is *identifying relationships*. The relationships between the aims and implications of research can be easily translated to the goals of applied settings, if the framework is shared by research and applied settings. For example, researchers might compile evidence which supports more restrictive regulation of advertising, and therefore find their work situated at the intersection of UEP strategies and the goal of regulation of industry. On the other hand, they might be unsure of the number of organizations who could advocate and raise awareness for their proposals. The mapping of campaign and advocacy groups at such intersections of the framework could therefore identify potential relationships between research and applied efforts and bolster their impact.

## The conceptual framework as a facilitator of communication and cooperation

Finally, the framework also facilitates *communication and cooperation*. Collating research and services on a common landscape highlights relationships between different areas of work and has the potential to cultivate productive interaction between stakeholders. A common framework facilitates a common language to discuss specific issues or initiatives. Furthermore, top-down organization can be facilitated through the mapping of provision to support the coordination of multiple stakeholder groups. For example, an organization focused on treatment and support for a specific group of individuals may wish to partner with other services to increase awareness of service availability. Having an easy-to-understand map of relevant organizations that offer similar provision in different sub-populations or regions, or a map of organizations whose focus is specifically education and awareness, could improve the impact of any action undertaken.

**Table 5. An example of the conceptual framework formatted as a matrix for mapping and evaluative purposes.**

| Socio-ecological Model | | Public Health Strategies | Public Health Goals | | | Gambling-Related Harms |
|---|---|---|---|---|---|---|
| | | | Regulation | Prevention | Support | |
| **Societal** | | **Education & Awareness** | | | | Resources |
| | | | | | | Relationships |
| | | | | | | Health |
| | | | | | | Non-specific |
| | | **Screening, Measurement & Intervention** | | | | Resources |
| | | | | | | Relationships |
| | | | | | | Health |
| | | | | | | Non-specific |
| | | **Environment & Product** | | | | Resources |
| | | | | | | Relationships |
| | | | | | | Health |
| | | | | | | Non-specific |
| **Community** | **Untargeted** | **Education & Awareness** | | | | Resources |
| | | | | | | Relationships |
| | | | | | | Health |
| | | | | | | Non-specific |
| | | **Screening, Measurement & Intervention** | | | | Resources |
| | | | | | | Relationships |
| | | | | | | Health |
| | | | | | | Non-specific |
| | | **Environment & Product** | | | | Resources |
| | | | | | | Relationships |
| | | | | | | Health |
| | | | | | | Non-specific |
| | **Targeted** | **Education & Awareness** | | | | Resources |
| | | | | | | Relationships |
| | | | | | | Health |
| | | | | | | Non-specific |
| | | **Screening, Measurement & Intervention** | | | | Resources |
| | | | | | | Relationships |
| | | | | | | Health |
| | | | | | | Non-specific |
| | | **Environment & Product** | | | | Resources |
| | | | | | | Relationships |
| | | | | | | Health |
| **Families and Social Networks** | | **Education & Awareness** | | | | Resources |
| | | | | | | Relationships |
| | | | | | | Health |
| | | | | | | Non-specific |
| | | **Screening, Measurement & Intervention** | | | | Resources |
| | | | | | | Relationships |
| | | | | | | Health |
| | | | | | | Non-specific |
| | | **Environment & Product** | | | | Resources |
| | | | | | | Relationships |
| | | | | | | Health |
| | | | | | | Non-specific |

*(Continued)*

**Table 5.** (Continued)

| Socio-ecological Model | Public Health Strategies | Public Health Goals | | | Gambling-Related Harms |
|---|---|---|---|---|---|
| | | Regulation | Prevention | Support | |
| **Individual** | **Education & Awareness** | | | | Resources |
| | | | | | Relationships |
| | | | | | Health |
| | | | | | Non-specific |
| | **Screening, Measurement & Intervention** | | | | Resources |
| | | | | | Relationships |
| | | | | | Health |
| | | | | | Non-specific |
| | **Environment & Product** | | | | Resources |
| | | | | | Relationships |
| | | | | | Health |
| | | | | | Non-specific |

## Conclusion

This paper has outlined our proposed conceptual framework for the prevention of GRH which was developed from the narrative analysis of findings from our scoping review. Our proposed framework aligns PH goals and strategies with existing harm frameworks and allows for differentiation at various 'levels' of society. By combining these features under a single overarching framework, stakeholders across disciplines can use a common language and work within a shared conceptual frame. The framework's utility is clearest as an applied tool that promotes the mapping of research, provision, or organizational focus. Doing so facilitates evaluative exercises which can identify important gaps in research and provision through an understanding of depth and breadth of coverage, and leading to informed decision-making and resource allocation. Moreover, mapping can identify relationships between work across disciplines and settings, which has potential to facilitate cross-sectoral communication and coordination. This could strengthen collective efforts, lead to the development of opportunities and initiatives, and encourage both research informed practice and stakeholder involvement in research.

There are four main considerations when critically evaluating the outcome of the current paper. Firstly, we acknowledge that other studies or reviews—depending on the sector under focus—may not have been returned under the search terms used within our scoping review, and there thus may be other approaches which have not been explored here. However, given the variety of sectors explored within our scoping review, we contend that the broad categorisations developed as the base of our collaborative framework would allow the inclusion of other approaches as part of any mapping exercises.

This links to the second consideration, which is that the categories that compose each strand of the framework are broad. However, as each strand (socio-ecological model; PH goals; PH strategies; GRH) is synthesised or adopted from current research, it is possible to dissect these components in greater detail by reviewing the appropriate literature. Thus, researchers *within* specific disciplines may wish to sub-categorise strands relevant to them. However, this broad categorization has been done intentionally and for pragmatic reasons. Our framework is designed to be highly interactable and for use across disciplines, sectors and settings. It is the intention that broad categories will encourage the framework to be a collaborative platform that is inclusive to all. As long as the intersections described in this paper remain at the heart of further refinements, the framework's utility as a tool for mapping, evaluation,

coordination and communication persists. Further sub-categorization within disciplines should only serve to demonstrate the flexibility of the framework and encourage greater and more nuanced understanding. Similarly, for applied settings, whether the full extent of non-academic activities (e.g., those activities of charities, financial institutions, advocacy groups or services) fall within the proposed public health strategies. Therefore, as the framework is intended to support stakeholders to reduce GRH, this initial proposition should act as a starting point for further refinement.

Thirdly, the distinctions between certain intersections of the conceptual framework are not always exclusive. Strategies can benefit more than one level of the social-ecological model, improve outcomes for more than a single harm or benefit, or can be both preventative and supportive from a PH perspective. The framework is not intended to place restrictions on classification and duplicating information across sections should not be seen as an issue. Although, we recommend that when this occurs in relation to mapping GRH, it would be useful to incorporate an understanding of harm taxonomies [5] when deciding where to best place a specific research paper.

Finally, the framework's conceptual utility is inherent in its depiction of how the component strands interact, and these relationships can be understood in greater detail as knowledge and research develops. The framework's use as a mapping tool for research and practice relies on an ongoing and systematic process of populating and updating information at the various intersections. This can be achieved by individual organizations and researchers if they are using the framework for a specific goal. Given that each individual organization or researcher will have a specific focus due to time constraints or speciality, there is also potential benefit from the development of an online application that can serve the whole community. Therefore, a future initiative could include the creation of such an application.

## Supporting information

**S1 Appendix. Grey literature.**
(DOCX)

**S1 Checklist. Preferred Reporting Items for Systematic reviews and Meta-Analyses extension for Scoping Reviews (PRISMA-ScR) checklist.**
(DOCX)

## Author Contributions

**Conceptualization:** Jamie Wheaton, Ben Ford, Agnes Nairn, Sharon Collard.

**Data curation:** Jamie Wheaton, Ben Ford.

**Formal analysis:** Jamie Wheaton, Ben Ford.

**Funding acquisition:** Agnes Nairn, Sharon Collard.

**Investigation:** Jamie Wheaton, Ben Ford.

**Methodology:** Jamie Wheaton, Ben Ford.

**Supervision:** Agnes Nairn, Sharon Collard.

**Validation:** Jamie Wheaton, Ben Ford.

**Writing – original draft:** Jamie Wheaton, Ben Ford.

**Writing – review & editing:** Agnes Nairn, Sharon Collard.

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
