## [Decision Letter · Decision Letter 0]

4 Sep 2023

PONE-D-23-21058Towards a conceptual framework for the prevention of gambling-related harms: Findings from a scoping reviewPLOS ONE

Dear Dr. Wheaton,

Thank you for submitting your manuscript to PLOS ONE. After careful consideration, we feel that it has merit but does not fully meet PLOS ONE’s publication criteria as it currently stands. Therefore, we invite you to submit a revised version of the manuscript that addresses the points raised during the review process.

We look forward to receiving your revised manuscript.

Kind regards,

Francis Xavier Kasujja

Academic Editor

PLOS ONE

Journal Requirements:

   "I have read the journal's policy and the authors of this manuscript have the following competing interests: JW, BF, AN, and SC have all received research funding from GambleAware. SC has received research funding from the Gambling Commission Regulatory Settlement funds. "

We note that you received funding from a commercial source:  GambleAware

Additional Editor Comments:

Please respond to all the issues raised by Reviewers 1 and 2.

Reviewers' comments:

Reviewer's Responses to Questions

**Comments to the Author**

1. Is the manuscript technically sound, and do the data support the conclusions?

Reviewer #1: Yes

Reviewer #2: Partly

2. Has the statistical analysis been performed appropriately and rigorously? 

Reviewer #1: Yes

Reviewer #2: N/A

3. Have the authors made all data underlying the findings in their manuscript fully available?

Reviewer #1: Yes

Reviewer #2: Yes

4. Is the manuscript presented in an intelligible fashion and written in standard English?

Reviewer #1: Yes

Reviewer #2: Yes

5. Review Comments to the Author

Reviewer #1: The authors have made an excellent job throughout the paper and in using systematic scoping review methodology. IThe paper is scientifically sound and strong, and I only have some minor notes the authors may want to consider.

Introduction clearly justify the need for this study. There is one excellent and potential paper to add when framing the study background: Price A, Hilbrecht M, Billi R. 2021. Charting a path towards a public health approach for gambling harm prevention. Z Gesundh Wiss. 29(1):37–53.

Research methods are described in sufficient detail and a scoping review methodology is a right choice here. One of the strengths of this review was that the search included grey literature, but on the other hand, their role, robustness and/or at least added benefits for this study were not discussed.

I was wondering that the search strategy begun from 2005 onwards and whether the same time cut-off was used when the authors searched grey literature. The authors cite UK-post legislation 2005. However, this choice of the year (2005) remains a bit unclear for me as a reader who is not familiar with the UK legislation.

I have no other comments. I wish the authors all the best with their important line of research.

Reviewer #2: Thank you for the possibility to review this interesting manuscript. The paper focuses on building a conceptual framework to assist in the prevention of gambling-related harms. Overall, such work is highly important. While gambling is increasingly understood as a public health issue, there has been a gap in our understanding regarding how the public health approach can be operationalised at different levels (treatment, prevention, policy). A conceptual framework that includes a wide understanding of harms and acknowledges the different actors in the field is both welcome and a necessary step forward.

However, having read the current manuscript, I feel that while the framework itself is relevant and its applicability is well described, there are some major shortcomings in how it was built methodologically.

The framework builds on a scoping review that appears to take different directions. The authors have reviewed existing work on frameworks on gambling harms (table 1) as well as public health strategies from gambling and other fields (table 2). Table 4 also uses some of the existing frameworks on gambling harms to build a summary.

How studies were chosen for each of these different sets of data and how they relate to each other is somewhat unclear. These should be described more systematically.

The identified public health strategies in gambling and other fields (table 2) also needs more elaboration. The included gambling literature is quite limited. Furthermore, there is no discussion on how gambling actually relates to or differs from the other included fields (alcohol, tobacco, HFSS). While some lessons could probably be learnt from these sectors, they also differ from gambling in many respects. Notably, digitalisation has impacted gambling in a very different way than these substance-based issues.

This leads me to question whether there could also be other public health strategies in the gambling field or whether the framework gives too much important to some strategies that are not so relevant for gambling.

The summary of harms (table 4) adopts the categorisation by Wardle et al. The justification that is given to this choice is that the categorisation is concise. However, is there a risk that such an approach can also neglect some important harm categories? The exclusion of crime is particularly unfortunate since as it stands, the model does not include harms that relate to the provision of gambling rather than the activity of gambling.

Minor issues:

The socio-ecological approach has been applied in a very similar way to gambling harms in the Wardle et al., framework. This should be acknowledged.

On p. 33 the authors note that they prefer language that is "less emotional" than for example "severity". Severity is a terms often used to describe the degree of gambling harms. I don't see how it is emotional.

There would be room for more concise language in many parts of the paper.

6. PLOS authors have the option to publish the peer review history of their article (what does this mean?). If published, this will include your full peer review and any attached files.

Reviewer #1: No

Reviewer #2: No

---

## [Author Response · Author response to Decision Letter 0]

11 Oct 2023

We would like to thank the reviewers for their timely and helpful feedback which has strengthened our paper. We are pleased that you feel our work is an important development in this area. Each of your comments required helpful amendments to the paper, and our individual responses can be found within the documentation submitted to PLOS One. Thank you again.

---

## [Decision Letter · Decision Letter 1]

15 Nov 2023

PONE-D-23-21058R1Towards a conceptual framework for the prevention of gambling-related harms: Findings from a scoping reviewPLOS ONE

Dear Dr. Wheaton,

Thank you for submitting your manuscript to PLOS ONE. After careful consideration, we feel that it has merit but does not fully meet PLOS ONE’s publication criteria as it currently stands. Therefore, we invite you to submit a revised version of the manuscript that addresses the points raised during the review process.

We look forward to receiving your revised manuscript.

Kind regards,

Francis Xavier Kasujja

Academic Editor

PLOS ONE

Journal Requirements:

Reviewers' comments:

Reviewer's Responses to Questions

**Comments to the Author**

1. If the authors have adequately addressed your comments raised in a previous round of review and you feel that this manuscript is now acceptable for publication, you may indicate that here to bypass the “Comments to the Author” section, enter your conflict of interest statement in the “Confidential to Editor” section, and submit your "Accept" recommendation.

Reviewer #2: (No Response)

Reviewer #3: All comments have been addressed

2. Is the manuscript technically sound, and do the data support the conclusions?

Reviewer #2: Yes

Reviewer #3: Yes

3. Has the statistical analysis been performed appropriately and rigorously? 

Reviewer #2: N/A

Reviewer #3: N/A

4. Have the authors made all data underlying the findings in their manuscript fully available?

Reviewer #2: Yes

Reviewer #3: Yes

5. Is the manuscript presented in an intelligible fashion and written in standard English?

Reviewer #2: Yes

Reviewer #3: Yes

6. Review Comments to the Author

Reviewer #2: Thank you again for this paper. I have re-read it with a lot of interest, and I think it has improved since the last version. The authors' responses were also very helpful to me as I was trying to understand some of the methodological choices made in the paper.

I still have some comments, that partly relate to my original comments.

First, Although you explained this to me well in the response, I think the article is still somewhat unclear about where the reviews included in table 1 (and 4) are coming from. I understand from your response to my previous comments that these tables were drawn separately from the systematic review. I think the confusion stems from the fact that this is a systematic review study, and such studies have a standard practice of reporting included studies in tables that look very much like table 1 (and 4).There is need for more clarity particularly on:

- How these frameworks were identified, based on what they were included in the table?

- What were the inclusion criteria for including some of these frameworks in table 4 later in the paper?

I think these issues could be explained under a separate heading in the methods section.

Second, I am still slightly stuck on the issue of crime. I appreciate the explanation given by the authors that crime can relate to the illegal provision of gambling, and was therefore excluded. This was mentioned at least in three different parts of the manuscript:

p.5.line 117; p. 10 line 136->; page 34. (version with track changes).

In the response to my previous comments, the authors note that crime that emerges from gambling is still included as a subcategory of harms. I think this needs to be in the manuscript as well. Furthermore, the paper by Banks & Waugh (that the authors also reference) also suggests that there are categories of gambling-related crime that are for example compliance-related. Are these included? I think just a clarifying sentence here would be enough.

Third, you note on page 11 line 173 (version with track changes) that you only based the analysis on peer reviewed literature and not grey literature. In the table with the included studies, you have also included some opinion pieces or correspondence. Are these peer reviewed?

Fourth, this is a suggestion: P. 31 has good discussion on the differences between gambling and substance-based commodities. You might add in a sentence or two summarising the comparison of the PH strategies identified across gambling/ other sectors. Were there strategies that were not captured by the gambling literature but that were included in the literature on the other commodities? Which ones were they and how did the emphasis differ across these products?

Reviewer #3: This is an excellent paper on a very important topic. This conceptual framework is highly needed in the sector as it will serve the needs of multiple stakeholders and support the development and implementation of various initiatives.

The authors have satisfactorily addressed the previous reviewers' comments. Only two small suggestions for consideration:

1. In the introduction, lines 35-36, the authors refer to "other harmful but legal sectors." Perhaps the authors want to reword so as to not label the other sectors as "harmful?" They instead may want to consider language such as "other sectors focused on reducing harms."

2. In a couple places the authors use the phrase "thanks to" in regards to digital transformation. They authors may want to consider "in part due to" instead.

7. PLOS authors have the option to publish the peer review history of their article (what does this mean?). If published, this will include your full peer review and any attached files.

Reviewer #2: No

Reviewer #3: No

---

## [Author Response · Author response to Decision Letter 1]

20 Dec 2023

Thank you to the reviewers for their helpful comments and feedback. Our response has been included within the attached submission.

---

## [Decision Letter · Decision Letter 2]

17 Jan 2024

Towards a conceptual framework for the prevention of gambling-related harms: Findings from a scoping review

PONE-D-23-21058R2

Dear Dr. Wheaton,

We’re pleased to inform you that your manuscript has been judged scientifically suitable for publication and will be formally accepted for publication once it meets all outstanding technical requirements.

Kind regards,

Francis Xavier Kasujja

Academic Editor

PLOS ONE

Additional Editor Comments (optional):

Reviewers' comments:

Reviewer's Responses to Questions

**Comments to the Author**

1. If the authors have adequately addressed your comments raised in a previous round of review and you feel that this manuscript is now acceptable for publication, you may indicate that here to bypass the “Comments to the Author” section, enter your conflict of interest statement in the “Confidential to Editor” section, and submit your "Accept" recommendation.

Reviewer #2: All comments have been addressed

2. Is the manuscript technically sound, and do the data support the conclusions?

Reviewer #2: Yes

3. Has the statistical analysis been performed appropriately and rigorously? 

Reviewer #2: N/A

4. Have the authors made all data underlying the findings in their manuscript fully available?

Reviewer #2: Yes

5. Is the manuscript presented in an intelligible fashion and written in standard English?

Reviewer #2: Yes

6. Review Comments to the Author

Reviewer #2: Thank you, you have an excellent job. The article is now very clear and a very important addition to the literature. I fully support its publication. Good luck with your future research!

7. PLOS authors have the option to publish the peer review history of their article (what does this mean?). If published, this will include your full peer review and any attached files.

Reviewer #2: No

---

## [Editor Report · Acceptance letter]

5 Mar 2024

PONE-D-23-21058R2 

PLOS ONE

Dear Dr. Wheaton, 

I'm pleased to inform you that your manuscript has been deemed suitable for publication in PLOS ONE. Congratulations! Your manuscript is now being handed over to our production team.

Kind regards, 

on behalf of

Dr. Francis Xavier Kasujja 

Academic Editor

PLOS ONE